# EDSSA: An Encoder-Decoder Semantic Segmentation Networks Accelerator on OpenCL-Based FPGA Platform

**DOI:** 10.3390/s20143969

**Published:** 2020-07-17

**Authors:** Hongzhi Huang, Yakun Wu, Mengqi Yu, Xuesong Shi, Fei Qiao, Li Luo, Qi Wei, Xinjun Liu

**Affiliations:** 1School of Electronic and Information Engineering, Beijing Jiaotong University, Beijing 100044, China; 17120009@bjtu.edu.cn (H.H.); 18120025@bjtu.edu.cn (Y.W.); lluo@bjtu.edu.cn (L.L.); 2Department of Electronic Engineering and BNRist, Tsinghua University, Beijing 100084, China; yumq17@mails.tsinghua.edu.cn; 3Intel Labs China, Beijing 100090, China; xuesong.shi@intel.com; 4Department of Precision Instrument, Tsinghua University, Beijing 100084, China; weiqi@tsinghua.edu.cn; 5Department of Mechanical Engineering, Tsinghua University, Beijing 100084, China; xinjunliu@mail.tsinghua.edu.cn

**Keywords:** FPGA, semantic segmentation, framework, OpenCL

## Abstract

Visual semantic segmentation, which is represented by the semantic segmentation network, has been widely used in many fields, such as intelligent robots, security, and autonomous driving. However, these Convolutional Neural Network (CNN)-based networks have high requirements for computing resources and programmability for hardware platforms. For embedded platforms and terminal devices in particular, Graphics Processing Unit (GPU)-based computing platforms cannot meet these requirements in terms of size and power consumption. In contrast, the Field Programmable Gate Array (FPGA)-based hardware system not only has flexible programmability and high embeddability, but can also meet lower power consumption requirements, which make it an appropriate solution for semantic segmentation on terminal devices. In this paper, we demonstrate EDSSA—an Encoder-Decoder semantic segmentation networks accelerator architecture which can be implemented with flexible parameter configurations and hardware resources on the FPGA platforms that support Open Computing Language (OpenCL) development. We introduce the related technologies, architecture design, algorithm optimization, and hardware implementation of the Encoder-Decoder semantic segmentation network SegNet as an example, and undertake a performance evaluation. Using an Intel Arria-10 GX1150 platform for evaluation, our work achieves a throughput higher than 432.8 GOP/s with power consumption of about 20 W, which is a 1.2× times improvement the energy-efficiency ratio compared to a high-performance GPU.

## 1. Introduction

Visual semantic segmentation is widely used in various applications, such as intelligent robot technology [1,2,3], autonomous driving [4,5], and pedestrian detection [6]. For intelligent robot technology, visual semantic SLAM (Simultaneous Localization and Mapping) that merges semantic information is a potential use of visual semantic segmentation for intelligent robot technology. It has been proven that the classic VSLAM (Visual SLAM) technology is an appropriate solution to the positioning and navigation of mobile robots [7,8], and that it can be implemented on low-power embedded platforms [9,10,11]. However, classic VSLAM is mostly based on low-level computer vision features (points, lines, etc.) when describing the surrounding environment. Although the description can extract geometric spatial information well, it lacks a high-level understanding of the environment in terms of semantics. In recent years, with the development of deep learning technologies, researchers have proposed various neural network algorithms to achieve high-level feature extraction based on computer vision technology, such as image classification [12,13,14] and semantic segmentation [15,16,17]. The semantic segmentation network based on a Convolutional Neural Network (CNN) has been widely implemented because of its high segmentation accuracy. The combination of classic VSLAM and a semantic segmentation network represents a new evolution of the traditional feature point extraction methods. Therefore, semantic VSLAM frameworks have been proposed to solve several problems with the classic VSLAM algorithms [18] and have shown good performance.

The architecture of the semantic segmentation network is mostly based on the CNN architecture. Using CNN architecture can not only achieve higher segmentation accuracy through network training, but is also suitable for many segmentation scenarios. However, it has several problems. First, the high segmentation accuracy rate usually means that the network is generally deep, which leads to a multiplied increase in the number of network parameters and calculations. This is reflected in higher requirements for the computing throughput of the hardware processing platform. Second, greater computing power and higher data transmission bandwidth often mean higher power consumption, which presents significant challenges to energy-constrained platforms. Furthermore, hardware platforms applied to mobile robots are limited in physical space. Therefore, the semantic VSLAM algorithm based on the semantic segmentation network must be able to run on a hardware platform with adequate computing capability and low power consumption, and be able to be embedded for use with mobile robots.

The hardware platform for processing of the SLAM algorithm on mobile robots mainly includes a CPU (Central Processing Unit) [11], FPGA [10,19], and ASIC (Application Specific Integrated Circuit) [9,20]. The platform for the semantic segmentation network is usually based on a GPU [15]. Semantic VSLAM using a semantic segmentation network is a topic of existing research. The CPU is irreplaceable as the logic processing core of the current hardware system. However, the computing power of the CPU appears to be limited in its ability to meet real-time requirements when implementing network computing. A GPU can provide a significant amount of computing power, but the higher power consumption cannot meet the needs of edge deployment. An ASIC has lower power consumption and smaller size but lower compatibility in terms of both software and hardware. An FPGA can provide higher computing power than a CPU, with lower power and a smaller volume than GPU. Furthermore, an FPGA cannot only be programmed with hardware description language (HDL) and IP (Internet Protocol) core development tools that focus on low-level hardware design and optimization, but can also be deployed with high-level design languages such as OpenCL tools. In addition, an FPGA also has higher compatibility with interfaces and hardware. These advantages make it suitable as the edge acceleration hardware for semantic segmentation networks. Therefore, a heterogeneous hardware platform may be a solution for intelligent mobile robots, including both the logic control and simple computing cores such as the CPU, and the heterogeneous acceleration hardware such as the FPGA.

In order to solve these problems, we propose an Encoder-Decoder semantic segmentation networks accelerator architecture (EDSSA) of an OpenCL-based FPGA heterogeneous platform. In the current study we test EDSSA with SegNet [16], a classic semantic segmentation network. The main points and contributions of this study are:(1)An FPGA hardware architecture based on OpenCL kernels was designed for Encoder-Decoder semantic segmentation network architecture. In this paper, we show the design details of the relevant architecture with the classic Encoder-Decoder semantic segmentation network SegNet as an example. The proposed architecture can also be applied to other Encoder-Decoder semantic segmentation networks by replacing the network models and OpenCL kernels.(2)We designed and explored the design space. The relationship between the design space and performance was explored on an Intel Arria-10 GX1150 platform to find the optimal solution. The proposed architecture can also be adapted to different hardware platforms using similar design space exploration methods.

The hardware acceleration of the SegNet inference process by EDSSA is shown. In Section 2, the network structure and the main mathematical operations of SegNet are discussed. In Section 3, the overall architecture of EDSSA and a series of instructions for the architectural design are given. In Section 4, we elaborate on the design and exploration of the design space. In Section 5, we introduce the measures of the optimization algorithm process and fixed-point quantization. In Section 6, the experimental platform used in this paper is provided, and we analyze the performance of the methods outlined in Section 3, Section 4 and Section 5. In the final section, we summarize this article.

## 2. Related Work

For a CNN accelerator based on FPGA, a series of studies have been undertaken. Chen Zhang et al. [21] explored the design space based on the roofline model and used the RTL design process to implement the classic CNN classification network based on FPGA. Mohammad Motamedi et al. [22] proposed an FPGA accelerator platform named PLACID that could generate an RTL-level architecture in Verilog. Huimin Li et al. [23] designed an accelerator for the classification network AlexNet. By optimizing calculation layer operations and design space exploration, the FPGA accelerator achieved a throughput up to 565.94 GOP/s with the Xilinx VC709. Although these studies have successfully developed an accelerator of the CNN network in FPGA, most use the RTL design method, which requires a significant time for development. Therefore, High-Level Synthesis (HLS) tools have become increasingly popular in both academic and industrial fields. Compared with the traditional methodology, HLS tools provide faster hardware development cycles and software-friendly program interfaces that can be easily integrated with user applications (e.g., PipeCNN). Based on the OpenCL developing tools, Jialiang Zhang et al. [24] solved the on-chip memory bandwidth limitation through the corresponding core design and implemented the inference process of Visual Geometry Group (VGG) on the Arria 10 GX1150 platform. Utku Aydonat et al. [25] implemented Winograd on the Arria 10 platform with OpenCL and achieved a throughput rate of up to 1382 GFLOPs. Dong Wang et al. [26] proposed a set of FPGA accelerators named PipeCNN which could be implemented on different FPGA platforms with reconfigurable performance and cost. In addition, various optimization methods have been proposed to achieve the FPGA accelerator design of the CNN network [27,28]. However, considering the difference in the algorithm flow and structure between the semantic segmentation network and the image classification network, several problems remain in the implementation of the FPGA accelerator of semantic segmentation networks:(1)The semantic segmentation networks usually contain an encoder and require computing layers such as unpooling or deconvolution;(2)Information feed-through between the decoder and the corresponding encoder exists in the semantic segmentation network;(3)The network may not contain the fully connected layer.(4)These problems are not addressed in previous research. Therefore, it is important to develop a semantic segmentation accelerator suitable for an FPGA platform based on HLS tools.

## 3. Description of Encoder-Decoder Semantic Segmentation Network

Compared with the traditional image classification network, the semantic segmentation network not only needs to identify and classify objects of a specific semantic category contained in the input image, but also needs to segment the geometric edges of the objects. Therefore, the semantic segmentation network has the following characteristics:(1)An end-to-end network. The input is an image and the output result is a segmentation label set with the same resolution as the input image, and the output of the image classification network is simply a number of category labels or probability values.(2)The network architecture includes both an encoder and decoder, whereas the image classification network only includes the encoder. The encoder is used to realize feature extraction, which often uses the classic image classification network as the filter. The decoder is used to realize semantic image restoration and obtain the semantic classification probability of each pixel.(3)The semantic segmentation network has data paths between the decoder and the corresponding encoder. In order to make up for the feature space information lost in the encoder process, the decoder usually introduces the features or pooled indexes generated by the encoder process to assist in completing the feature recovery.

Figure 1 shows the network architecture of the SegNet-A classic Encoder-Decoder semantic segmentation network. The input image passes through the encoder of the network for feature extraction and generates the corresponding pooling indices in the pooling layers. Then, the extracted features are used for feature restoration through the decoder. The main functions and mathematical calculations of each calculation layer of SegNet are introduced in Code 1, and the relevant parameters are shown in Table 1.

Convolutional layer. The convolutional layer is the main computing layer in the CNN model. Its main function is feature extraction. Usually, the input of the convolutional layer is a number of feature maps. These feature maps and the corresponding convolution kernels perform two-dimensional convolution operations to extract local features. Then, the results between different feature maps are summed. After adding the bias, a local feature description value corresponding to a convolution kernel is generated. Different local features are extracted by sliding the two-dimensional convolutional windows on the input feature map, and the output high-dimensional feature map is finally generated. In this process, the convolution kernels used to generate an output feature map are shared, and the number of convolution kernels determines the number of output feature maps. In addition, the Batch Normalization (BN) layer [29] and the Rectified Linear Unit (RELU) layer are connected after each convolution layer in SegNet.

Pooling layer. The pooling layer is usually located after the CONV layer. It aims to reduce the amount of calculation and control overfitting. The pooling operation is applied to each input feature map separately. This means that the input and output of the pooling layer have the same number of feature maps. The operations between different feature maps are independent of each other. In the SegNet model, maximum pooling is used.

Unpooling layer. The unpooling layer is the inverse operation of the pooling layer. The unpooling layers output the feature maps with the same resolution as the corresponding pooling layer according to the pooling index address. Each unpooling operation places the input feature at the position corresponding to the pooled index address and fills other positions with 0. Similarly, the unpooling operation is applied to each input feature map independently.

The computing characteristics of each computing layer in SegNet also determine the strategy for hardware implementation on the FPGA. For the convolutional layers, we can see that the main operation of Figure 2 is multiply–accumulate. Moreover, the convolution operation is independent between different input feature maps (different C_cin_) and different convolution kernels (different N_k_). Such an operation structure is highly suitable for parallel computing acceleration. The main operations of the pooling and unpooling layers are comparison and reorder, so it is suitable for designing an efficient pipeline to accelerate the operations. Considering that operations in the pooling layers and the unpooling layers are independent between different feature maps, parallel multi-threading can be used for acceleration.

## 4. Overall Architecture Design

Here, we first introduce the FPGA development process based on OpenCL, which can be divided into two parts: the host and the device. The Host mainly runs OpenCL-based context and command queue management and controls all memory data transmission and kernel execution queues. Generally, users need to build the host programming code that is complied with the OpenCL development specification to call the corresponding OpenCL API (Application Programming Interface) to control the devices. The device side, or FPGA board, is mainly used for kernels execution and pipeline control. The user should get the FPGA executing image that is finally used for FPGA by undertaking code building, FPGA compilation and synthesis, and simulation and debugging of the kernels in the OpenCL development environment. The image can be used to configure the FPGA to deploy the kernels and corresponding component functions.

To implement the deployment of SegNet in FPGA, EDSSA uses the overall architecture shown in Figure 3 based on OpenCL. The function execution is mainly realized by the kernels on-chip. The data storage is divided into two parts: on-chip memory and off-chip memory. These are used to store the features and parameters required or generated at different stages of the kernel execution process.

### 4.1. Configurable Pipes and Layer Connections

EDSSA realizes different layer connection modes through configurable data flow pipes and layer connections, which realizes non-blocking data flow between the kernels. If using off-chip global memory as a reference, a sub-process for reading, calculating, and storing feature data can be described as:

off-chip global memory (input features, parameters, or pooling indices) **→** on-chip cache buffers **→** convolution kernel **→** data pipes **→** pooling or unpooling kernel (if needed) **→** data pipes **→** on-chip cache buffers **→** off-chip global memory (output features or pooling indices).

In order to adapt to the structure of SegNet shown in Figure 1, there are four kinds of sub-process modes designed to configure data flow pipes and layer connections:C_F = 00: off-chip global memory (input features and parameters) → on-chip cache buffers **→** convolution kernel **→** data pipes **→** on-chip cache buffers **→** off-chip global memory (output features);C_F = 01: off-chip global memory (input features and parameters) **→** on-chip cache buffers **→** convolution kernel **→** data pipes **→** pooling kernel **→** data pipes **→** on-chip cache buffers **→** off-chip global memory (output features and pooling indices);C_F = 10: off-chip global memory (input features, parameters, and pooling indices) **→** on-chip cache buffers **→** convolution kernel **→** data pipes **→** unpooling kernel **→** data pipes **→** on-chip cache buffers **→** off-chip global memory (output features);C_F = 11: off-chip global memory (input features, parameters, and pooling indices) **→** on-chip cache buffers **→** convolution kernel **→** data pipes **→** pooling kernel **→** data pipes **→** unpooling kernel **→** data pipes **→** on-chip cache buffers **→** off-chip global memory (output features).

These four modes are controlled by Data Flow Controller Flag (C_F) to configure the kernel to be executed and select the data pipes for data transmission. The sub-processes of these four modes share the same cache, data transmission component, and convolution kernel, and the difference is whether the convolution kernel is connected to the pooling or unpooling kernel and the data pipes used for data transmission. The entire network structure of SegNet can be realized through the combination of these four modes of sub-processes. If C_F is used to represent the sub-process mode, the combination of sub-processes that implement SegNet is: 00-01-00-01-00-00-01-00-00-01-00-00-11-00-00-10- 00-00-10-00-00-10-00-10-00-00.

The framework of EDSSA has the advantages as follows: (a) When executing each sub-process, we can ensure that each core is executed at most once, so as to ensure that there is no contention for the same kernel hardware, thus, ensuring that the entire sub-process is not blocked. (b) The same hardware component will be used when performing the same kernel function in different sub-processes. It reduces the hardware resource overhead on the FPGA chip. (c) Adoption of the FIFO (First Input First Output)-based pipe design means all data is transmitted on-chip during a sub-process, which greatly reduces the transmission delay and improves the overall throughput rate. (d) Only a simple 2-bit control word C_F can control all modes of the sub-process.

### 4.2. Kernels Design

EDSSA has three OpenCL kernels for completing the three calculation layers of SegNet: the convolution kernel, pooling kernel, and unpooling kernel. The convolution kernel contains all of the functional components and computing units required to implement the convolutional, BN, and RELU layers. The pooling kernel and unpooling kernel complete all of the computing units required by the pooling layer and unpooling layer, respectively.

Convolution kernel. As shown in Figure 4a, the core of the convolution kernel is a three-dimensional array of multiply–accumulate units, which contains C × N × K_c_/4 units. Each unit is completed by a 4-input 8-bit × 8-bit high-efficiency multiply–accumulate MAC (Multiply Accumulate) IP core. The input of this array is the input features and weights of the corresponding two-dimensional convolution operation, and the output result is the partial sum. The parallelism of the array calculation depends on C × N. A higher C × N means a higher calculation throughput rate and a higher calculation and transmission cost. The parallel accumulators and shift register groups are connected behind the array, which is used to buffer the partial sum, and finally outputs the complete sum. Then, the output values go through operations such as quantization, accumulating bias, and RELU. Finally, the output leaves the kernel through the data pipes selected by the control word C_F and is transferred to the next stage.

Pooled kernel. As shown in Figure 4b, the core of the pooling kernel is a set of efficient pipelines based on register sets. The input features are imported by the data pipes and then compared with the corresponding feature stored in the row register. Then, the bigger one is compared with the feature stored in the column register until the largest feature value in the pooling window is obtained. Finally, the output leaves the kernel through the data pipes selected by the control word C_F and transferred to the next stage. Considering that the pooling operation is independent between the different feature maps, multi-threaded pipelines are used to improve the core throughput with the parallelism as N.

Unpooling kernel. As shown in Figure 4c, the core of the unpooling kernel is a set of efficient pipelines based on register sets with different clock domains. A line register set based on a ping-pong operation is designed to achieve feature filling and output at the same time. The input features and the corresponding pooling indices are imported by the data pipes with a 4× clock domain. The features will be stored in the line registers with a 2× clock domain by the corresponding address according to the value of the pooling indices. The remainder of the registers corresponding to other addresses of the unpooling window will be filled with 0. At the same time, another set of line registers that has been filled will export the output features in the new maps with a 1× clock domain. Finally, the output will leave the kernel through the data pipes. As for the pooling kernel, the unpooling kernel also uses multi-threaded pipelines with a parallelism of N.

### 4.3. Memory Access Design

Due to the large number of features and parameters during SegNet processing, it is impossible to store all data on-chip during each sub-pipeline process. Therefore, EDSSA stores the feature maps and parameters of each sub-process in the off-chip large-capacity global memory. At the beginning of each sub-process, the memory access controller reads a part of the input features and parameters into the on-chip cache RAM (Random Access Memory) according to the designed reading mode and then transmits it into the kernel through the data pipes. The off-chip memory adopts the ping-pong design to store the input and output feature maps separately, which aims to improve the system throughput rate. In addition, the parameters stored by the on-chip RAM will be shared in the convolution kernel to calculate different output characteristics. It can reduce the delay caused by data transmission. We also use the vectorized data structures for data storage and transmission to ensure that more features and parameters can be transmitted into the array at the same time in a calculation cycle, which leads to a higher system throughput rate. The vectorization dimension mainly depends on C and N.

## 5. Design Space Exploration and Optimization

### 5.1. Design Space Exploration

The purpose of design space exploration is to balance the performance and hardware resource consumption of the FPGA accelerator. In EDSSA, the design space exploration is implemented by changing the value of C and N. These two parameters not only determine the throughput rate of the multiply–accumulate array but also affect the data structure of the input and output features and the number of threads in pooling and unpooling kernels. In addition, a higher value of C and N means a higher data vectorization dimension with more resource consumption. EDSSA adopts the vector structure shown in Figure 5 for features, weights, and bias. The size of the parameters C and N is equal to the parallelism C and N of the multiply–accumulate array. Each time the multiply–accumulate array is operated, the controller first copies N identical C-dimensional vectorized features to obtain the C × N-dimensional vectorized features, and then the C × N-dimensional vectorized features and the C × N-dimensional vectorized weights are imported into the array to be computed. Finally, an N-dimensional vectorized output is obtained. The feature vector will adopt an N-dimensional structure when quantizing, pooling, and unpooling.

When exploring the impact of parallelism on the system design space, we use the parameter declarations in Table 1. First, we assume that all kernels of the system use the same clock unit with the multiply–accumulate array to complete a calculation. No blocking exists in the system pipeline in one clock cycle. The array can complete all of the C × N multiply–accumulate operations within each clock. In this assumption, we know that the number of clocks corresponding to completing a sub-process can be expressed as:(1)Ncompute=Wcout∗Hcout∗Nk∗(Kc∗Kc∗Ccin+1)

Then the number of clocks required is:(2)$$Nclk=\frac{Ncompute}{C*N}=\frac{Wcout*Hcout*Nk*\left(Kc*Kc*Ccin+1\right)}{C*N}$$

Thus, increasing C × N can reduce the clock cycle required by the sub-process. However, simply increasing C or N may not achieve better utilization of hardware resources. The effect of increasing C on the design space is:(a)Increasing the number of multiply-accumulate units;(b)Increasing the vectorized data bit width of the input features and weights.(c)For N:(a)Increasing the multiply-accumulate unit;(b)Increasing the vectorized data bit width of the weights and bias;(c)Increasing the number of threads in pooling and unpooling kernel.(d)Therefore, increasing C or N may result in:(a)Double the consumption of computing logic resources;(b)Higher data transmission and storage costs;(c)Higher timing requirements to meet data synchronization.

In order to achieve the optimal hardware utilization with the specific FPGA device, it is necessary to find the most suitable C and N by design space exploration.

SegNet has an excellent performance in semantic segmentation applications with the well-designed network architecture for a full set of processes including training and inference. However, for network deployment on edge devices, it is more important to implement efficient inference implementation. Some algorithm processes that focus on achieving better performance in the training process can be optimized during inference. Therefore, EDSSA optimizes the algorithm flow and quantizes the data for reducing the complexity of the algorithm and hardware overhead while maintaining a certain accuracy of segmentation.

### 5.2. Algorithm Flow Optimization

EDSSA, like most CNN accelerators, focuses on solving the acceleration of the inference process of neural networks on FPGA terminals. Therefore, EDSSA discards the softmax layer, merges convolution and BN operations, and uses relative pooling indices addresses instead of the global one. These steps are described as follows:

Discard the softmax layer: The softmax layer is discarded for the following reasons. First, the mathematical function of the softmax is a kind of normalization algorithm to count the segmentation probability of the output pixels. It does not change the statistical results of the output feature map. Second, the softmax layer is used only once in the algorithm. Therefore, it is wasteful to sacrifice precious on-chip computing resources to realize the softmax layer, considering that EDSSA is based on the OpenCL that can reasonably allocate and manage command execution on the host and devices. Therefore, we abandon the on-chip deployment of the softmax layer and deploy it to the host for implementation.

Merging convolution and BN operations: The mathematical operations of convolution operations (Formula (3)) and BN operations (Formula (4)) are both multiply–accumulate. Therefore, EDSSA simplifies the algorithm flow by merging convolution and BN operations (Formula (5)). Through parameter preprocessing, α·WEIGHTS and α·BIAS+β in Equation (5) can be regarded as two new parameters equivalent to $WEIGHTS^{\prime }$ and $BIAS^{\prime }$, and participate in the convolution operation. Merging convolution and BN has the following benefits:(a)Simplify the algorithm flow while retaining the accuracy of the calculation results;(b)Reduce the number of pipeline stages and save the hardware overhead required for BN operations;(c)Reduce the number of quantization and the system quantization accuracy loss.
(3)$$OUTPUT\_conv=\underset{\left(Kc,\;Kc\right)}{\sum }WEIGHTS\cdot INPUT+BIAS$$
(4)OUTPUT_bn=α·OUTPUT+β
(5)$$output\_bn=\underset{\left(Kc,\;Kc\right)}{\sum }\left(\alpha \cdot WEIGHTS\right)\cdot INPUT+\left(\alpha \cdot BIAS+\beta \right)$$

Using relative pooling index addresses: SegNet uses 32-bit floating-point global addresses to store the corresponding pooling index in the caffe [30]-based training and inference. On the one hand, 32-bit floating-point addresses use more hardware resources for transmission and storage. On the other hand, the feature map space information contained in the global address is redundant for the unpooling process. The spatial information of a specific feature in a feature map is correlated with the number of pipeline clock cycles of the unpooling kernel. This means once we have the relative position of the unpooling window in the specified cycle, the unpooling kernel can place the feature correctly. Therefore, EDSSA uses 2-bit fixed-point relative pooled index addresses as shown in Figure 6. Thus, a significant amount of storage space is saved for indices addresses, while simplifying the hardware overhead of address generation.

### 5.3. Quantization

The purpose of fixed-point quantization is to compress the bit width, and reduce the hardware resource costs of data calculation and transmission. However, fixed-point quantization and lowering the bit width will cause a loss of calculation accuracy. In the worst case, it may lead to erroneous calculation results. Considering that the SegNet network is a computational and storage-intensive algorithm, a suitable fixed-point quantization strategy can significantly reduce hardware resource consumption and increase system processing speed. In EDSSA, we perform N-bit fixed-point linear quantization on all the features and parameters [31]. The quantization can be described by Formulas (6) to (9).

(a)Arrange input $x_{i}$ in absolute value, and find the maximum:(6)$$\left|Max\right|=max\left(abs\left(x_{i}\right)\right)$$(b)Get the fractional bit:
(7)$$f_{x_{i}}=ceil\left({log}_{2}\frac{\left|Max\right|}{2^{N-1}-1}\right)$$(c)For each $x_{i}$ element of input, set:(8)$$x_{i}^{\prime }=round\left(x_{i}*2^{-f_{x_{i}}}\right)$$(d)Bit truncation. Limit $x_{i}^{\prime }$ to N bits:(9)$$x_{j}^{\prime }=\{\begin{array}{c} 2^{M-1}-1,\;x_{j}^{\prime }>2^{M-1}-1\;\\ -2^{M-1},x_{j}^{\prime }<-2^{M-1}\\ x_{j}^{\prime },-2^{M-1}\leq x_{j}^{\prime }\leq 2^{M-1} \end{array}$$

In addition, EDSSA performs a dynamic M-bit fixed-point quantization on each output result of the multiply–accumulate array because multiplication doubles the bit width of the data, which means that for different convolutional layers and different feature maps in one convolutional layer, the quantization bit width is N bits, but the fractional bit is different. The purpose of using dynamic fixed-point quantization is to reduce accuracy loss. The quantization is represented by $x_{j}={\left(-1\right)}^{s}\cdot \left(\sum_{i=0}^{M-2}2^{i}\cdot m_{i}\right)\cdot 2^{-f_{j}}$, where *S* is the sign bit, *M* is the quantization bit width, $m_{i}$ is the mantissa, and $f_{j}$ is the fractional bit. $f_{j}$ can be obtained by the network training process.

## 6. Results

We used the development tool based on an Intel FPGA SDK for OpenCL pro 17.1 to implement the development of EDSSA. The hardware platform is HERO [32], a heterogeneous platform that can be deployed on medium-sized robots. The host uses a CPU system based on an Intel i5-7260U, and the device uses an FPGA board based on an Intel Arria-10 GX1150 connected with the host by PCIE 3.0 x8. The SegNet model is trained based on the PASCAL VOC 2012 dataset [33], and dynamic fixed-point quantization is performed. The input image is an RGB image with a resolution of 224 × 224. The calculation methods of throughput and energy efficiency are given in [34].

### 6.1. Quantization

In order to determine the optimal quantization strategy, we explored the effect of different quantization bit widths *M* on algorithm accuracy. In caffe, 32-bit floating-point data is used for network training and inference, and the final global accuracy, class accuracy, and mIoU(Mean Intersection over Union) are 82.80%, 62.30%, and 46.30%, respectively. Based on this model, we used the proposed quantization strategy in the inference process of SegNet-Basic [16] with the data set of CamVid at 480 × 360 resolution, and the results obtained are given in Figure 7. We can see that when the data is quantified with a bit width less than 16 bits, the quantization error starts to appear and increases as the bit width decreases. Without the dynamic quantization strategy, class accuracy and mIoU decrease significantly when the bit width is less than 12 bits. However, the trend of accuracy declines after using dynamic quantization has obviously eased. This shows the necessity of the dynamic fixed-point quantization strategy in the low-bit width quantization. In addition, even if dynamic quantization is performed, when the bit width is lower than 8 bits, the three accuracies are greatly reduced in value. The quantization accuracy losses of global accuracy, class accuracy, and mIoU are 3.82%, 6.30%, and 4.78%, respectively.

In summary, the quantization strategy used in EDSSA is 8-bit dynamic fixed-point quantization. At this time, the quantization accuracy losses of global accuracy, class accuracy, and mIoU in SegNet inference in the test set of PASCAL VOC 2012 are 0.8%, 1.1%, and 1.6%, respectively.

### 6.2. Runtime Performance

The main factor affecting the runtime of EDSSA is the design space parameters C and N. Figure 8a shows the runtime of EDSSA with different C and N. When C × N is higher, the running speed of system will be faster. This shows that a higher degree of computation and thread parallelism has a direct effect on the speed of the accelerator. Moreover, the running speed increases exponentially when using a lower degree of parallelism. However, for a high degree of parallelism, the speed improvement achieved slows down, and there may even be no gains (such as C × N = 16 × 32 and 16 × 64). This means that there are other factors that restrict the system speed. One of these is the clock frequency of the kernels. Figure 8b shows the kernel clock frequency of EDSSA for different C and N. When a higher degree of parallelism C × N is used, the kernel clock frequency tends to decrease. The higher the parallelism, the more obvious the drop in clock frequency. This may be because the higher parallelism means higher data transmission timing requirements and more pipeline threads, which may reduce the system clock. In addition, when using the same C × N, choosing a larger C can achieve a faster running speed. This means that increasing C (mainly to increase the bit width of the vectorized data of the input features and weight parameters) compared to increasing N (mainly to increase the bit width of the vectorized data of the weight parameters and the number of threads in the pipeline) has a smaller effect on reducing the speed of the system.

### 6.3. Hardware Resource Consumption

In order to explore the impact of design space on EDSSA hardware resource consumption, we tested the DSP (Digital Signal Processing), RAM blocks, and logic utilization of FPGA cores under different C and N. The relevant results are given in Figure 8c–e. It can be seen from the experimental results that the DSP utilization rate is the same when using the same parallelism; if the parallelism is doubled, the DSP consumption is also doubled. The results prove that DSP is mainly used to generate multiply–accumulate unit arrays, and a higher C × N will exponentially increase the resource consumption of computing components. In addition, for the Arria-10 GX1150 platform, when the parallelism C × N = 1024, the DSP utilization rate reaches 100%, which means that FPGA resources cannot support higher parallelism. Analysis of RAM occupancy and logic utilization data shows that higher C × N consumes more on-chip storage and logic resources. In addition, choosing a larger C under the same C × N requires fewer resources, indicating that the resources occupied by increasing the number of threads for pipelines are higher than the transmission and storage consumption of increasing the bit width of vectored data.

### 6.4. Throughput and Energy Efficiency Assessment

Through the evaluation of runtime and resource consumption, we obtained the best design space under the Arria-10 GX1150 platform with a degree of parallelism C × N = 32 × 32. The test results can be summarized in Table 2 and the output semantic segmentation results can be seen in Figure 9. Under the optimal design, we used all DSP resources to implement the multiply–accumulate array, while consuming 63% of on-chip RAM blocks and 24% of the logic resources, and finally achieved a system throughput rate of 432.8 GOP/s at the kernel clock frequency of 202 MHZ. Table 3 shows the comparison with other platforms with SegNet.

## 7. Conclusions

In the future, semantic SLAM based on semantic segmentation network will be the key technology for intelligent mobile robots to achieve autonomous motion. Considering that the hardware resources of the embedded platform are limited, the solution of accelerating the semantic segmentation network in the edge devices has become a top priority. In this paper, we show EDSSA, an accelerator framework for semantic segmentation networks, which can be implemented with flexible parameter configurations and hardware resources on the FPGA platforms that support OpenCL development. EDSSA achieved a system throughput of 432.8 GOP/s and about 16.65 GOP/J based on the Intel Arria-10 GX1150 platform.

## Figures and Tables

**Figure 1 sensors-20-03969-f001:**
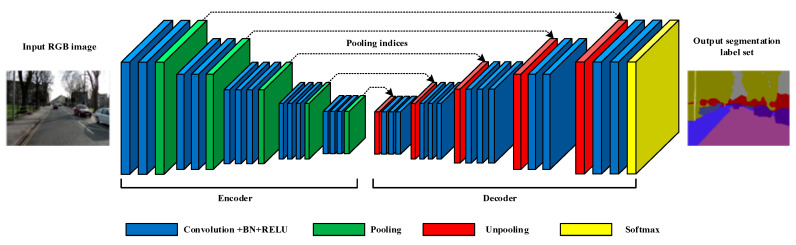
SegNet architecture [16].

**Figure 2 sensors-20-03969-f002:**
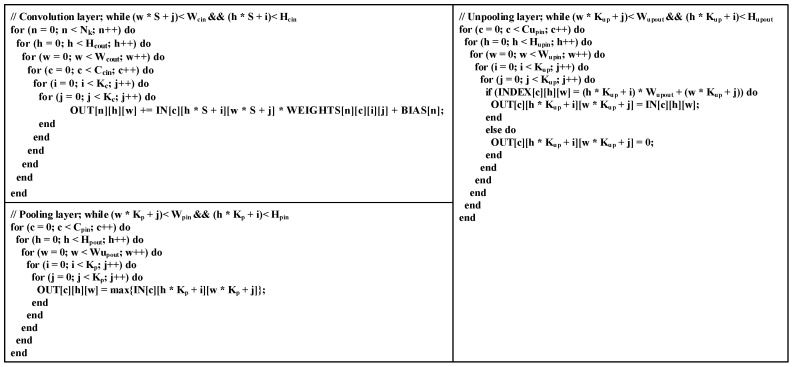
Code description of convolutional, pooling, and unpooling layers.

**Figure 3 sensors-20-03969-f003:**
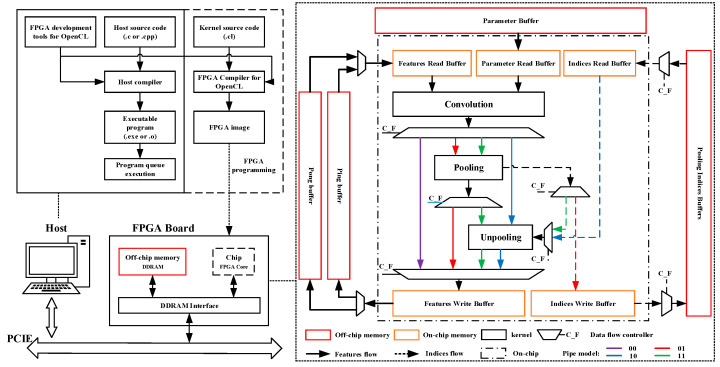
FPGA development process for OpenCL and Encoder-Decoder semantic segmentation networks accelerator (EDSSA) overall architecture.

**Figure 4 sensors-20-03969-f004:**
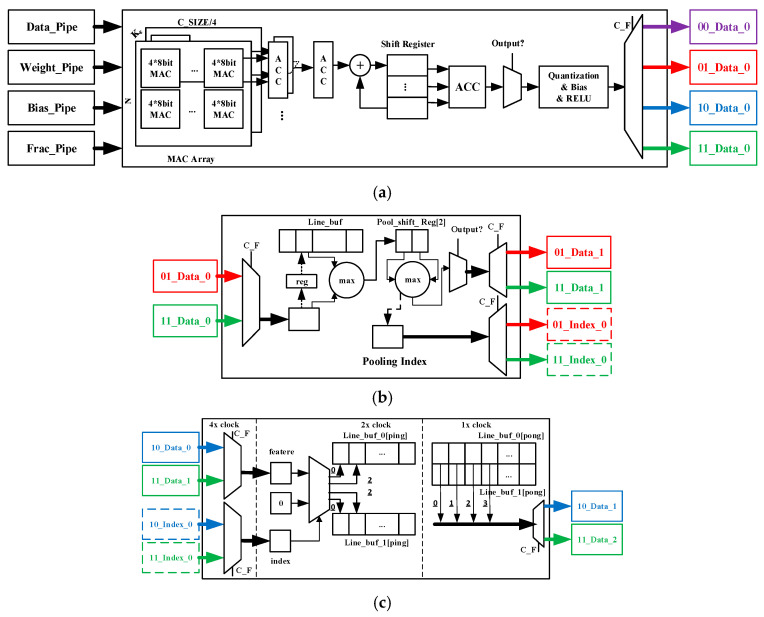
Architecture of kernels in EDSSA. (**a**) Convolution kernel; (**b**) pooling kernel; (**c**) unpooling kernel.

**Figure 5 sensors-20-03969-f005:**
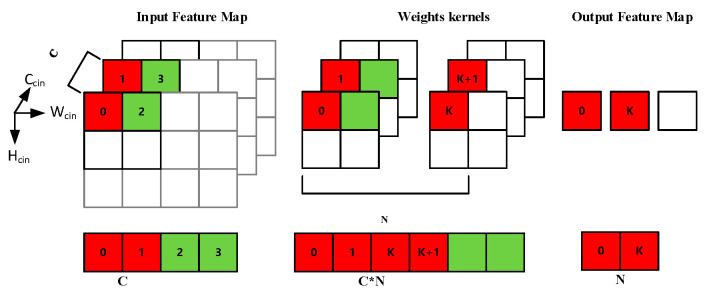
Vectorized data structure.

**Figure 6 sensors-20-03969-f006:**
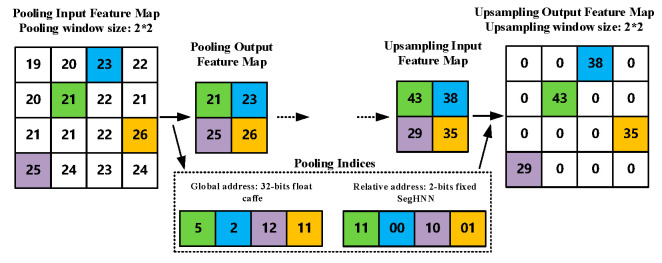
Global and Relative Pooling indices.

**Figure 7 sensors-20-03969-f007:**
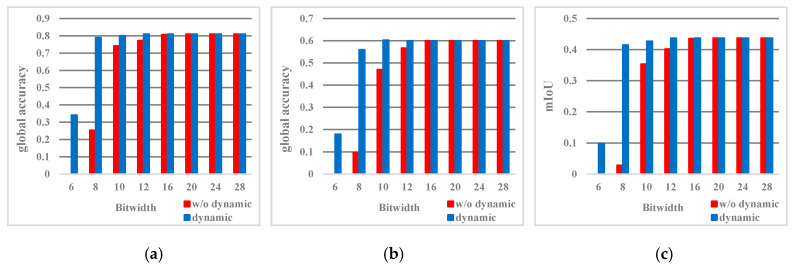
Accuracy with different quantization strategy. (**a**) global accuracy; (**b**) class accuracy; (**c**) mIoU.

**Figure 8 sensors-20-03969-f008:**
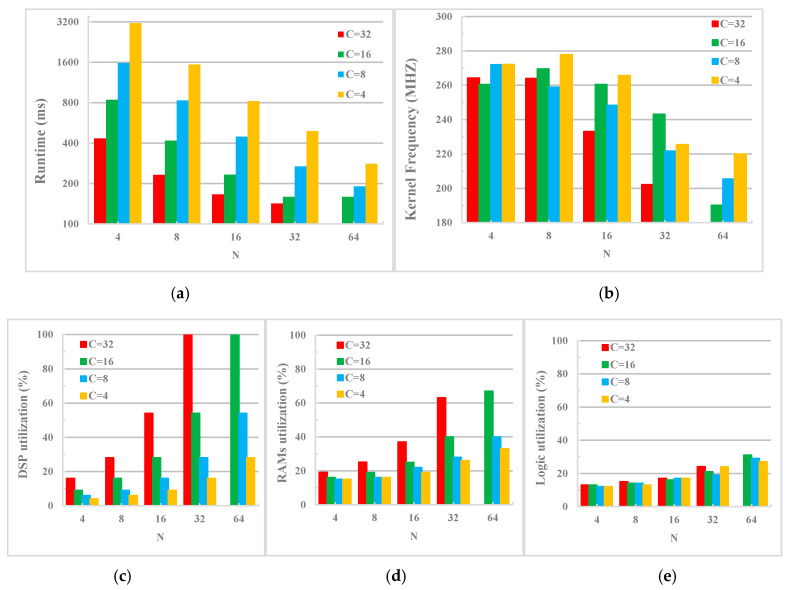
Design space exploration results for EDSSA on the Arria-10 GX1150. (**a**) Runtime; (**b**) Frequency; (**c**) DSP utilization; (**d**) RAMs utilization; (**e**) Logic utilization.

**Figure 9 sensors-20-03969-f009:**
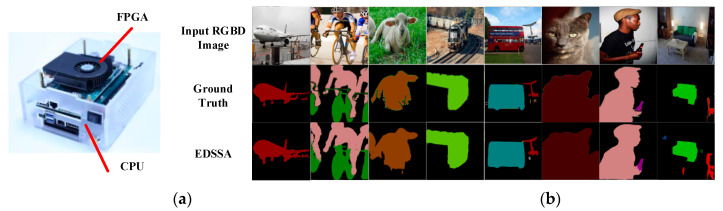
Platform and semantic segmentation result samples. (**a**) HERO [32]; (**b**) result.

**Table 1 sensors-20-03969-t001:** Parameters in convolutional, pooling, and unpooling layers.

Convolution Parameters	Description	Pooling Parameters	Description	Unpooling Parameters	Description
W_cin_	Width of the input feature maps	W_pin_	Width of the input feature maps	W_upin_	Width of the input feature maps
H_cin_	Height of the input feature maps				
W_cout_	Width of the output feature maps	H_pin_	Height of the input feature maps	H_upin_	Height of the input feature maps
H_cout_	Height of the output feature maps				
C_cin_	Numbers of input feature maps	C_pin_	Numbers of input feature maps	C_upin_	Numbers of input feature maps
P_c_	Padding sizes	W_pout_	Width of the output feature maps	W_upout_	Width of the output feature maps
K_c_	Size of the convolution kernel				
N_k_	Numbers of convolution kernel	H_pout_	Height of the output feature maps	H_upout_	Height of the output feature maps
S_c_	Sliding step				
BIAS	Bias	K_p_	Size of the pooling windows	K_up_	Size of the pooling windows

**Table 2 sensors-20-03969-t002:** Summary of EDSSA with best parallelism.

Device	Resource Capacity	Resource Consumed	Runtime (ms)	Kernel Frequency (MHz)	System Throughput (GOP/s)
Arria-10 GX1150	Logic 427,200	Logic 101,955 (24%)	141.8	202.08	432.8
	RAM blocks 2713	RAM blocks 1703 (63%)			
	DSP blocks 1518	DSP blocks 1515 (100%)			

**Table 3 sensors-20-03969-t003:** Comparison with other platforms with SegNet.

Platform	Devices	Typical Platform Power (W)	Accelerator Power (W)	Throughput (GOP/s)	Energy-Efficiency (GOP/J)
CPU only	Intel E3-1230 V2	70	69	19.0	0.28
CPU+GPU	Intel E3-1230 V2 & Nvidia GTX1080	70 + 180	173	2397.8	13.86
CPU+FPGA	HERO (Intel Core i5 7260U & Arria-10 GX1150)	15 + 25	26	432.8	16.65

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
