# Peer review of "EDSSA: An Encoder-Decoder Semantic Segmentation Networks Accelerator on OpenCL-Based FPGA Platform"

_sensors, 2020, doi:10.3390/s20143969_

Round 1

Reviewer 1 Report

The paper is looking good. I advise at least put some photos of the hardware and experiments done. 

Can you describe in more detail the target applications?

I find that graphics could be better described. It's easy to read for someone that knows this subject.

Why not you convert from RGB to Grayscale and do comparisons? To handle in Grayscale requires less power processing of computer system.

In terms of the target application for autonomous driving, we can note that is important we define the ROI - Region of Interest.

Reviewer 2 Report

Firstly, the design work developed is remarkable,
even when an HLS based on OPENCL has been used.
However, this article persists in a dispute that is difficult
to justify if the comparison data is not provided accurately.

I am not going to discuss if the available devices
in the comparison are adequate
(as representatives of different technologies such as multi-cpu,
GPGPU and FPGAs);
but I am not convinced to establish a comparison based on GOP / s
or GOP / s / w when it is not about mathematical operations
with the same resolution and when the consumption data provided is
not clarified how they have been obtained. The 20 watt data is not
credible if it is not clarified how this data has been obtained.

And also the 250 watt data from the GPGPU should be endorsed with the experiment that determined that value.

It gives the impression as if some catalog data had been used and little else.

I do not quite understand, and since there is a heterogeneous computing platform (HERO), as the comparison does not establish, starting from the same Opencl, of the same solution implemented only with the I5, i5 with GPU or i5 with Arria 10.

That way it would at least justify the implementation work and I think it could justify the possible acceptance of the article (although I am very afraid that it will be discovered and have to publish that the most energy efficient solution, and therefore important for ROBOT, not is to work with the FPGA).

I have seen many comparisons in the DNN world trying to establish FPGAs as winners, always at the cost of limited resolution. I think that this article does not contribute anything decisive in that line and of course it has been left with a lot of data to contribute.

Finally, the inference data and the loss of accuracy when the resolution has been reduced should be showed in greater detail. Were training samples or validation samples used to show the loss of accuracy found in Figure 6?

How does it affect the generalizability of the network?

I reiterate again that the implementation work has left a very good impression on me and I think it can be an acceptable job with the recommendations that I have included, even if they are not the results of the comparison they do not turn out as one expects.

Round 2

Reviewer 2 Report

It has become clearer to me how the energy efficiency ratings of both CPU and CPU-GPU solutions have been done.
And I also appreciate the explanations as to why the same CPU was not used in all experiments. Now it seems clearer which platform was used.

With this, the third table has been clear to me in how it has been obtained, by the answer of the authors, the values obtained for the cases belonging to the first and second row (CPU and CPU+GPU), especially in what I was more worried about: the fourth column (Accelerator Power)
However I would need to know in detail how they have obtained the value of Accelerator Power of the third case (CPU+FPGA): It is by means of a multimeter, or it is by means of a software monitoring or it is an estimation of the compiler of the Opencl kit of intel?
